

# Varying intensities of chronic stress induce inconsistent responses in weight and plasma metabolites in house sparrows (*Passer domesticus*)

Ursula K. Beattie[1], Nina Fefferman[2] and L. Michael Romero[1]

[1] Department of Biology, Tufts University, Medford, Massachusetts, United States
[2] Departments of Ecology and Evolution, University of Tennessee—Knoxville, Knoxville, Tennessee, United States

Corresponding author
Ursula K. Beattie,
ursula.beattie@tufts.edu

## ABSTRACT

One of the biggest unanswered questions in the field of stress physiology is whether variation in chronic stress intensity will produce proportional (a gradient or graded) physiological response. We were specifically interested in the timing of the entrance into homeostatic overload, or the start of chronic stress symptoms. To attempt to fill this knowledge gap we split 40 captive house sparrows (*Passer domesticus*) into four groups (high stress, medium stress, low stress, and a captivity-only control) and subjected them to six bouts of chronic stress over a 6-month period. We varied the number of stressors/day and the length of each individual bout with the goal of producing groups that would experience different magnitudes of wear-and-tear. To evaluate the impact of chronic stress, at the start and end of each stress bout we measured body weight and three plasma metabolites (glucose, ketones, and uric acid) in both a fasted and fed state. All metrics showed significant differences across treatment groups, with the high stress group most frequently showing the greatest changes. However, the changes did not produce a consistent profile that matched the different chronic stress intensities. We also took samples after a prolonged recovery period of 6 weeks after the chronic stressors ended. The only group difference that persisted after 6 weeks was weight—all differences across groups in metabolites recovered. The results indicate that common blood metabolites are sensitive to stressors and may show signs of wear-and-tear, but are not reliable indicators of the intensity of long-term chronic stress. Furthermore, regulatory mechanisms are robust enough to recover within 6 weeks post-stress.

## INTRODUCTION

Despite decades of research, the transition from acute to chronic stress is still poorly understood. The reactive scope model (*Romero, Dickens & Cyr, 2009*) is a theoretical framework designed to understand this transition. The model posits that detrimental chronic stress symptoms could develop because of accumulated damage (wear-and-tear), with greater damage eliciting an earlier transition. A computational model of reactive

scope (J Wright, K Buch, UK Beattle, PMG Gormally, LM Romero, N Fefferman, 2023, unpublished data) has highlighted gaps in investigations into the effect of chronic stress intensity and suggested points of interrogation that might further elucidate this transition. The current experiments are part of a larger project in which chronic stress intensity was manipulated with the hope of varying the accumulation of damage and thus the likelihood of developing chronic stress symptoms (*Beattie et al., 2023*).

The reactive scope model (*Romero, Dickens & Cyr, 2009*) describes four potential concentrations of physiological mediators in relation to the stress response; (1) Levels in the predictive homeostasis range exist to respond to daily or seasonal changes; (2) levels in the reactive homeostasis range are needed to respond to unpredictable changes, or acute stressors; (3) homeostatic overload occurs in ranges in which the physiological mediator itself becomes harmful; (4) homeostatic failure occurs in ranges of the physiological mediator which are too low to sustain life. Predictive and reactive homeostasis are levels in which the mediator promotes survival and are thus beneficial. Homeostatic overload and homeostatic failure represent times in which the mediator (or lack thereof) causes pathology and is thus harmful. The entrance into a chronic stress state can thus be modeled as the transition from reactive homeostasis into homeostatic overload and wear-and-tear can be modeled as a decrease of this threshold.

One of the most-commonly studied physiological mediators is corticosterone, the primary glucocorticoid in birds. At both baseline and stress-induced concentrations, corticosterone exerts major effects on metabolism. These metabolic effects are mediated through the high-affinity low-capacity mineralocorticoid receptor at baseline levels (*Blas, 2015*; *Romero & Wingfield, 2016*) and through the low-affinity high-capacity glucocorticoid receptor during periods of acute stress (*Landys et al., 2004*). The metabolic effects of corticosterone aid survival during acute stress (*Wingfield et al., 1998*) by acting permissively on catecholamine function to convert glycogen to glucose (*Romero & Wingfield, 2016*). During chronic stress, corticosterone is important in promoting the long-term conversion of energy stores to glucose to aid in the recovery of a stressor (*Romero & Beattie, 2021*). In the context of the reactive scope model, corticosterone in the predictive homeostasis range promotes normal foraging for basal energetic needs and storing energy for future use. In the reactive homeostasis range, corticosterone promotes energy mobilization to survive or recover from a stressor. During homeostatic overload and failure, corticosterone causes metabolic dysregulation, which can manifest as vital tissue breakdown or diseases such as diabetes (*Dallman & Bhatnagar, 2001*; *Korte et al., 2005*; *Romero, Dickens & Cyr, 2009*).

Throughout this stress experiment, we assessed three blood metabolites in wild-caught house sparrows (*Passer domesticus*). We measured glucose, ketones (specifically, β-hydroxybutyrate), and uric acid, during both fed and fasted states. Although glucose is tightly regulated in birds (*Alonso-Alvarez & Ferrer, 2001*; *Basile et al., 2021*; *Castellini & Rea, 1992*; *Rodríguez, Tortosa & Villafuerte, 2005*), one primary effect of corticosterone during acute stress is to raise plasma glucose levels (*Davies et al., 2013*; *Deviche et al., 2016b*) (but see *Deviche et al., 2016a*, *2014*; *Fokidis et al., 2011*), potentially to aid in the recovery of the stressor or to prepare for the next stressor (*Munck & Koritz, 1962*; *Romero*

& Wingfield, 2016). Ketones spike in the bloodstream when fat deposits are broken down, and can be used as an indication of fasting (Berg, Tymoczko & Stryer, 2002; Buyse & Decuypere, 2015; Cherel et al., 1988; Totzke et al., 1999), short-term mass change (Cerasale & Guglielmo, 2006), and as proxy for performance (Kaliński et al., 2022; Lindholm, Altimiras & Lees, 2018). Corticosterone can increase plasma ketones as a result of gluconeogenesis during chronic stress (Bernard et al., 2002; Viblanc et al., 2018). Uric acid is produced as a byproduct of protein breakdown and serves as a major antioxidant in birds (Castellini & Rea, 1992; Scanes, 2015a). Corticosterone also breaks down proteins for gluconeogenesis during chronic stress (Scanes, 2015a; Viblanc et al., 2018). Additionally, the birds in our lab eat a diet that consists primarily of sunflower seeds, so uric acid levels (in a fed-state) can also be used as a proxy for food consumption (Beattie et al., 2022; Hafez, Abdel-Rahman & Naguib, 2017).

Energy usage and metabolism have been studied within the scope of chronic stress before, but with variable results. Most of these studies use a period of chronic stress on the order of days to weeks (Bauer et al., 2022; Cyr et al., 2007; Gormally et al., 2018, 2019b; Neuman-Lee et al., 2015; Xie et al., 2015), which are occasionally followed by a recovery period (Beattie et al., 2022), and even more rarely, a second period of chronic stress (Awerman & Romero, 2010; Gormally et al., 2019a). In the present experiment, we attempted to manipulate wear-and-tear by gradually increasing, decreasing, or maintaining chronic stress intensity over a prolonged period of time. If chronic stress can create a gradient of responses, like acute stress does (Armario, Marti & Gil, 1990; Armario et al., 1986; Armario, Montero & Balasch, 1986; Nephew, Kahn & Romero, 2003; Rich & Romero, 2005; Terman, Morgan & Liebeskind, 1986), and this effect manifests in metabolic changes, we would expect to see differences in metabolite levels across treatment groups differing in stressor intensity. More specifically, if the onset of stress-related pathology is sensitive to (and indicative of) stressor intensity, we would expect to see an increase in fasting uric acid (indicating vital protein breakdown and thus, homeostatic overload (Romero, Dickens & Cyr, 2009)) earlier in more highly-stressed individuals.

## MATERIALS AND METHODS

### Experimental design

A description of the experimental design for this large project are presented in Beattie et al. (2023) and repeated briefly here. We caught 40 wild house sparrows (17M:23F) using mist nets in suburban backyards of Eastern Massachusetts in mid-June of 2021. These birds acclimate well to captivity and are commonly used in physiological stress experiments (Hanson et al., 2020). Birds were doubly housed (M/F or F/F pairs) in cages (45 cm × 37 cm × 33 cm), put on a light cycle of 12L:12D, and allowed at least 3 weeks to acclimate to captivity before experiments began. Birds occasionally got pumice stones in their cage as enrichment. During the acclimation period, birds were not disturbed except for routine husbandry. Birds were given ad libitum access to food (commercially available mix of millet and sunflower seeds) and water during the acclimation period and throughout the experiment. At any point during this experiment, if a bird dropped below 85% of its pre-experiment (taken after at least 3 weeks of acclimation) weight or showed any signs of
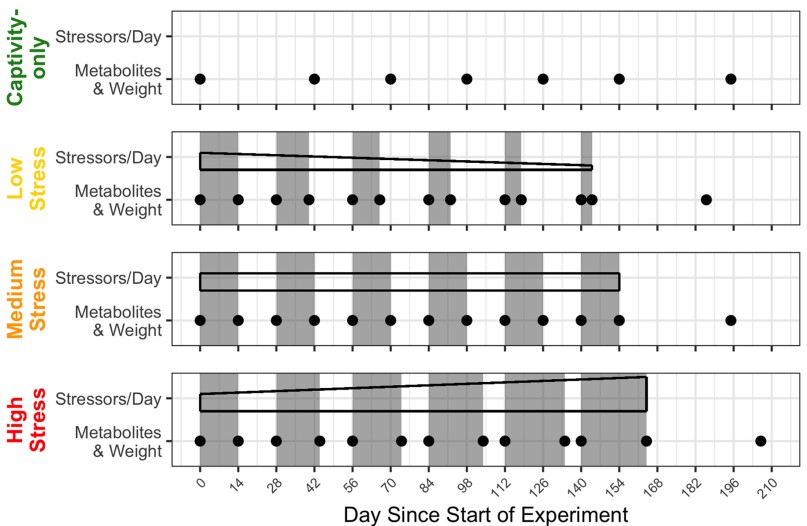

**Figure 1 Timeline of chronic stress and samples taken.** Panels correspond to treatment group, with group names on the left side. Grey rectangles indicate periods of applied chronic stress. Black-outlined shapes represent a decrease, consistent, or increase in the numbers of stressors birds experience in day. Black dots indicate a blood sample taken.

declining health, the bird was removed from the experiment and allowed to recover. If its health/weight recovered, the bird was allowed back into the experiment. During this long-term experiment, some birds died for unknown reasons and so sample sizes are not equal across groups and time (Table S1). A Cox survival analysis (*Cox, 1972*; *Therneau, 2022*) showed no statistical significance of sample size across treatment groups ($z = 0.26$, $p = 0.79$) (*Beattie et al., 2023*). At the end of the experiment, remaining birds were used for other experiments in the lab (not part of the present study).

Birds were randomly sorted into four experimental groups: captivity-only controls (5M:5F), low stress (4M:6F), medium stress (4M:6F), and high stress (5M:5F). A power analysis indicated that we would have a 95% chance of detecting a 50% difference in means with a sample size of eight birds in each group. We chose a sample size of 10 birds/group because this was such a long-term study and we anticipated some birds dying (see above). All chronic stress treatments were applied concurrently and all birds were sampled in random order. The experimenters were not blinded to the treatment groups during stressor application or sampling, but were during metabolite analysis.

A true control group was not possible because housing wild birds in captivity is inherently stressful. We compensated for the lack of a control group by employing a repeated-measures design and by using a "captivity-only control" group that experienced only the stress of captivity, routine husbandry, and minimal blood sampling. The medium stress group repeated 2 weeks of four stressors/day, followed by 2 weeks of a break for 6 months. The low stress and high stress groups started at 2 weeks of four stressors/day and 2 weeks of a break, but each subsequent bout of stress either decreased or increased by 2 days in length and gradually decreased or increased the number of stressors/day, respectively. After 6 months, all groups were given a prolonged recovery period of 6 weeks, at which one

final blood sample was taken. A timeline of the stress regimens and timepoints of blood sampling can be found in Fig. 1 (*Beattie et al., 2023*).

The chronic stress protocol was adapted from an established protocol that involves rotating acute psychological stressors (*Cyr et al., 2007*; *Rich & Romero, 2005*). Stressors were randomly selected and applied for 30 min at randomly selected times during the day. The stressors were: restraint in a cloth bag, cage rolling (rolling the cage racks around the room), cage tapping (tapping the cages with a pen), playing a radio in the bird room, human speaking in the bird room, placing wind-up toys on the cage floor, running high speed fans in the direction of the cages, and flashing colorful lights (always done during the dark period). The captivity-only treatment group was housed separately from the three chronic stress groups. At times when certain groups needed stressors but others did not, the group was removed and the stressor was performed in a third room (*Beattie et al., 2023*).

All animals were collected under a Massachusetts state collection permit and all experiments were approved by the Tufts University Institutional Animal Care and Use Committee and were performed according to the guidelines for the use of wild birds in research (*Fair et al., 2010*).

## Metabolite analysis

At the beginning and end of every bout (Fig. 1), we took small blood samples for metabolite analysis. A total of 14 h before sampling (approximately 1 h prior to lights-off the night before), food dishes were removed from the cages and fresh cage liners were placed to eliminate the possibility of the birds eating any dropped food during the fast. Within 3 min of entering the room, approximately 10 µL of blood were taken by puncturing the alar vein with a 26-gauge needle and collecting blood in a heparinized capillary tube. The birds were then given food *ad libitum*, allowed to feed undisturbed, and 1 h later another 10 µL blood sample was taken in the same manner. The following day, we measured each bird's weight with a Pesola spring scale to the nearest 0.5 g.

Immediately after blood samples were taken, we measured glucose, β-hydroxybutyrate (a ketone), and uric acid on whole blood using previously validated point-of-care devices (*Beattie et al., 2022*; *Lindholm, Altimiras & Lees, 2018*; *Morales et al., 2020*; *Stoot et al., 2014*). The point-of-care devices used were the Precision Xtra Blood Glucose and Ketone Monitoring System (Cat. No. 98814-65) with the glucose (Cat. No. 99838-65) and ketone (Cat. No. 70745-BX) test strips and the UASure Blood Uric Acid Meter (Cat. No. U3003). Approximately 1 µL was needed for the glucose and ketone devices and approximately 2 µL was needed for the uric acid device.

*Beattie et al. (2022)* had previously validated a similar sampling regimen and metabolite analysis in house sparrows; however, we performed an additional validation to ensure that the changes we detected could be attributed to feeding. In a separate cohort of captive house sparrows (*n* = 6), we quantified glucose, β-hydroxybutyrate, and uric acid in birds that had their food reintroduced remotely. Birds were fasted for approximately 14 h, however instead of removing the food from the cage, the food dishes were covered with a lid that could be opened by pulling a string from outside of the room (*Fischer, Franco &*

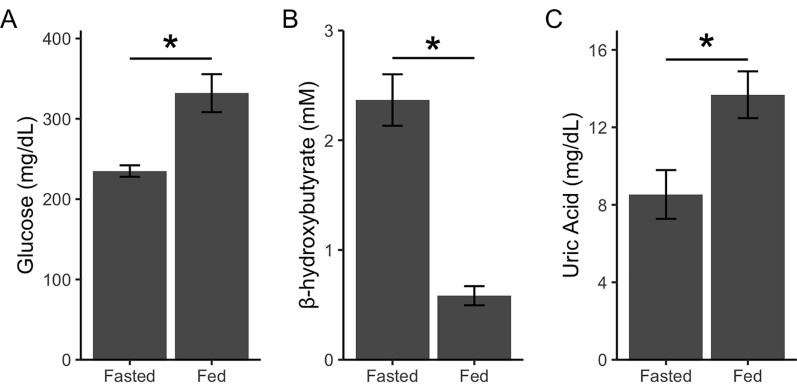

**Figure 2 Point-of-care devices detect post-prandial metabolite changes.** Glucose and uric acid increase after feeding because of glucose and uric acid in diet (A and C). β-hydroxybutyrate decreases (B) after feeding as birds stop relying on ketones for energy (as during phase II of fasting). Refeeding was done remotely to show that experimenter presence during refeeding does not affect metabolite levels. Asterisks indicate $p < 0.05$.                     

*Romero, 2016*). Birds were then bled 1 h later and metabolites were quantified as described above. We remotely fed the birds in the same manner twice before taking the blood samples so birds would habituate to any potential neophobia responses to the dish cover (*de Bruijn & Romero, 2019*). Fasted blood samples were taken on a separate day. Data was analyzed using linear mixed models with bird identity as a random effect and state (fed/fasted) as a fixed effect. Glucose ($F_{1,5} = 14.95$, $p = 0.01$; Fig. 2A) and uric acid ($F_{1,5} = 9.65$, $p = 0.02$; Fig. 2C) significantly increased after feeding, while β-hydroxybutyrate significantly decreased after feeding ($F_{1,5} = 50.70$, $p = 0.001$; Fig. 2B). Thus, remotely refeeding and refeeding in person result in the same pattern of change in plasma metabolites (*Beattie et al., 2022*). These data allowed us to take fasted and fed blood samples on the same day during the main experiment and attribute changes after feeding to the feeding itself and not the acute stress of an experimenter replacing the food.

For the main experiment, in instances where the value was higher than the detection limit of the device, a ceiling value of 500 and 20 mg/dL was assigned for glucose and uric acid, respectively. A total of 31 out of 783 samples exceeded the uric acid test, one out of 783 samples exceeded the glucose test, and no samples exceeded the test for ketones. To determine outliers, we averaged all values across the experiment for fasted glucose, fed glucose, fasted ketones, fed ketones, fasted uric acid, and fed ketones. Any values that fell outside of the range of the mean ± 3 standard deviations were excluded from further analysis. Two samples were removed for fasted glucose, nine samples were removed for fasted ketones, and five samples were removed for fed ketones.

For each of the three metabolites, we divided each fed value by its respective fasted value to calculate a postprandial fold-change. Whereas the absolute values for the metrics in a fasted and fed state gave insight into the long-term impact of chronic stress on baseline values (fasted) and concentrations after a meal (fed), the postprandial fold-change revealed whether metabolites after a meal increased from baseline consistently over time.
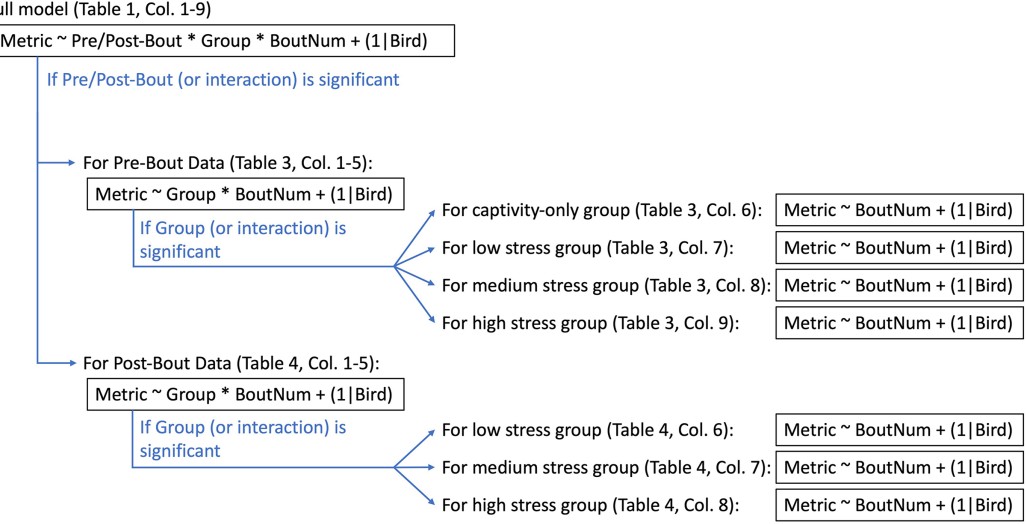

**Figure 3 Statistical pipeline for weight and metabolite analyses.** "Metric" (to the left of the tilde) indicate what metric is being statistically tested (*i.e.*, weight, fasted/fed/fold-change glucose, ketones, or uric acid). The factors to the right of the tilde are the factors being tested in the statistical analysis. The asterisk indicates where interactions were tested. "(1|Bird)" indicates that bird was a random effect and thus controlled for statistically. Captivity-only birds were only sampled at "pre-bout" timepoints (though they did not actually experience bouts of stress).

## Statistics

All statistical analyses were run in R version 4.0.3 (*R Core Team, 2020*). We used linear mixed effect models ('lmer' function, lme4 package (*Bates et al., 2015*)) with treatment group, bout number, and pre/post-bout (samples taken at the beginning of the bout were pre-bout, samples taken at the end of the bout were post-bout) as fixed effects and bird identity as a random effect ("full model"; Fig. 3). We then ran a Type III ANOVA ('Anova' function, car package (*Fox & Weisberg, 2011*)). In the case of a significant effect of pre/post-bout or a significant interaction with pre/post-bout, we split the data into a pre-bout subset and a post-bout subset and re-ran the models with treatment group and bout number as fixed effects and bird identity as a random effect ("pre-bout data" and "post-bout data"; Fig. 3). In the case of a significant effect of treatment group or an interaction with treatment group, we split the groups into four separate subsets, and re-ran the model with bout number as a fixed effect and bird identity as a random effect. A flow chart of this statistical pipeline can be found in Fig. 3. Despite the multiple iterations of models to determine effects of bout, the most important statistical results we were interested in were the effects of treatment and interactions with treatment.

We used a standard ANOVA to compare across groups in the recovery samples (those taken 6 weeks after the experiment concluded). The medium stress group was excluded from the recovery analysis because there were only three birds left. For metrics that had significant group effects at the recovery timepoint, we calculated an eta$^2$ effect size using the 'eta_squared' function in the effectsize package (*Ben-Shachar, Lüdecke & Makowski, 2020*).

**Table 1 Results from the main data analysis models.**

| (1) Metric | (2) | (3) Group | (4) Pre/Post-Bout | (5) BoutNum | (6) Group * Pre/Post-Bout | (7) Group * BoutNum | (8) Pre/Post-Bout * BoutNum | (9) Group * Pre/Post-Bout * BoutNum | (10) Group (at 6 weeks recovery) |
|---|---|---|---|---|---|---|---|---|---|
| Weight | | $F_{3,123} = 5.95$ $p = 8.04 \times 10^{-4}$ | $F_{1,290} = 4.46$ $p = 0.04$ | $F_{5,294} = 2.42$ $p = 0.04$ | $F_{2,291} = 2.69$ $p = 0.07$ | $F_{15,292} = 1.72$ $p = 0.05$ | $F_{5,291} = 1.65$ $p = 0.15$ | $F_{10,291} = 2.06$ $p = 0.03$ | $F_2 = 4.06$ $p = 0.04$ |
| Glucose | Fasted | $F_{3,299} = 0.97$ $p = 0.41$ | $F_{1,296} = 3.54$ $p = 0.06$ | $F_{5,299} = 4.10$ $p = 1.29 \times 10^{-3}$ | $F_{2,295} = 0.63$ $p = 0.53$ | $F_{5,296} = 5.68$ $p = 0.01$ | $F_{5,296} = 5.68$ $p = 5.04 \times 10^{-5}$ | $F_{10,296} = 3.13$ $p = 7.89 \times 10^{-4}$ | $F_2 = 0.78$ $p = 0.48$ |
| | Fed | $F_{3,256} = 2.33$ $p = 0.07$ | $F_{1,295} = 1.57$ $p = 0.21$ | $F_{5,298} = 1.49$ $p = 0.19$ | $F_{2,295} = 0.31$ $p = 0.73$ | $F_{15,298} = 1.03$ $p = 0.42$ | $F_{5,295} = 3.61$ $p = 3.51 \times 10^{-3}$ | $F_{10,295} = 3.22$ $p = 5.86 \times 10^{-4}$ | $F_2 = 1.51$ $p = 0.25$ |
| | Fold-change | $F_{3,326} = 4.17$ $p = 6.45 \times 10^{-3}$ | $F_{1,300} = 6.86$ $p = 9.27 \times 10^{-3}$ | $F_{5,298} = 1.21$ $p = 0.30$ | $F_{2,297} = 0.98$ $p = 0.38$ | $F_{15,298} = 1.24$ $p = 0.24$ | $F_{5,297} = 1.72$ $p = 0.13$ | $F_{10,296} = 1.18$ $p = 0.30$ | $F_2 = 1.05$ $p = 0.37$ |
| Ketones | Fasted | $F_{1,230} = 0.11$ $p = 0.96$ | $F_{1,286} = 0.37$ $p = 0.54$ | $F_{5,291} = 1.01$ $p = 0.41$ | $F_{2,286} = 0.71$ $p = 0.49$ | $F_{15,298} = 1.61$ $p = 0.07$ | $F_{5,286} = 2.77$ $p = 0.02$ | $F_{10,287} = 3.58$ $p = 1.71 \times 10^{-4}$ | $F_2 = 0.59$ $p = 0.56$ |
| | Fed | $F_{3,284} = 3.60$ $p = 0.01$ | $F_{1,291} = 8.27$ $p = 4.32 \times 10^{-3}$ | $F_{5,296} = 9.52$ $p = 1.97 \times 10^{-8}$ | $F_{2,291} = 2.26$ $p = 0.10$ | $F_{15,295} = 2.67$ $p = 7.68 \times 10^{-4}$ | $F_{5,291} = 4.55$ $p = 5.18 \times 10^{-4}$ | $F_{10,291} = 2.58$ $p = 5.21 \times 10^{-3}$ | $F_2 = 0.94$ $p = 0.41$ |
| | Fold-change | $F_{2,292} = 1.86$ $p = 0.14$ | $F_{1,283} = 5.56$ $p = 0.02$ | $F_{5,289} = 5.48$ $p = 7.75 \times 10^{-5}$ | $F_{2,283} = 2.41$ $p = 0.09$ | $F_{12,287} = 2.29$ $p = 4.48 \times 10^{-3}$ | $F_{5,283} = 6.80$ $p = 5.23 \times 10^{-6}$ | $F_{10,284} = 4.47$ $p = 7.26 \times 10^{-6}$ | $F_2 = 1.46$ $p = 0.26$ |
| Uric Acid | Fasted | $F_{3,269} = 3.76$ $p = 0.01$ | $F_{1,296} = 0.49$ $p = 0.48$ | $F_{5,300} = 2.19$ $p = 0.06$ | $F_{2,296} = 0.47$ $p = 0.62$ | $F_{15,299} = 2.25$ $p = 0.07$ | $F_{5,297} = 2.25$ $p = 0.05$ | $F_{10,297} = 2.59$ $p = 5.00 \times 10^{-3}$ | $F_2 = 0.48$ $p = 0.62$ |
| | Fed | $F_{3,187} = 1.49$ $p = 0.22$ | $F_{1,275} = 3.12$ $p = 0.08$ | $F_{5,298} = 1.89$ $p = 0.10$ | $F_{2,295} = 3.71$ $p = 0.03$ | $F_{15,297} = 1.30$ $p = 0.20$ | $F_{5,295} = 1.47$ $p = 0.20$ | $F_{10,295} = 1.49$ $p = 0.14$ | $F_2 = 0.60$ $p = 0.56$ |
| | Fold-change | $F_{3,329} = 3.51$ $p = 0.02$ | $F_{1,297} = 1.89$ $p = 0.17$ | $F_{5,301} = 3.96$ $p = 1.70 \times 10^{-3}$ | $F_{2,298} = 1.31$ $p = 0.27$ | $F_{15,301} = 1.49$ $p = 0.11$ | $F_{5,298} = 0.80$ $p = 0.55$ | $F_{10,298} = 1.86$ $p = 0.05$ | $F_2 = 0.68$ $p = 0.52$ |

**Note:**

Columns 3–9 show the results of the linear mixed-effect model, "metric~group * pre/post-bout * bout number", with bird identity as a random effect. The metrics are listed in columns 1–2, group is treatment group (captivity-only, low stress, medium stress, and high stress), pre/post-bout refers to when the sample was taken (pre-bout or post-bout), and bout num is the bout number (1–6 from Fig. 1). Column 10 reflects the results of the ANOVA on the samples taken 6 weeks after the experiment ended, where the model is "metric~group". Cells that are bolded contain p-values less than 0.05.

For each step of the analysis (see Fig. 3), we verified that the models did not violate the assumption of homogeneity of variances through Levene's Test and by visually inspecting the residual plots. For steps of the analysis with datasets that did not pass Levene's test, we used Tukey's Ladder of Powers to transform the data. The following datasets were log transformed: fed ketones (pre- and post-bout together), fed ketones (pre-bout subset), fold-change ketones (pre- and post-bout together), fold-change ketones (pre-bout subset), fold-change ketones (pre-bout, captivity-only group subset). Fasted ketones (pre-bout, low stress group subset) was raised to the power of −3. The reciprocal root was taken of fed ketones (pre-bout, captivity-only group subset).

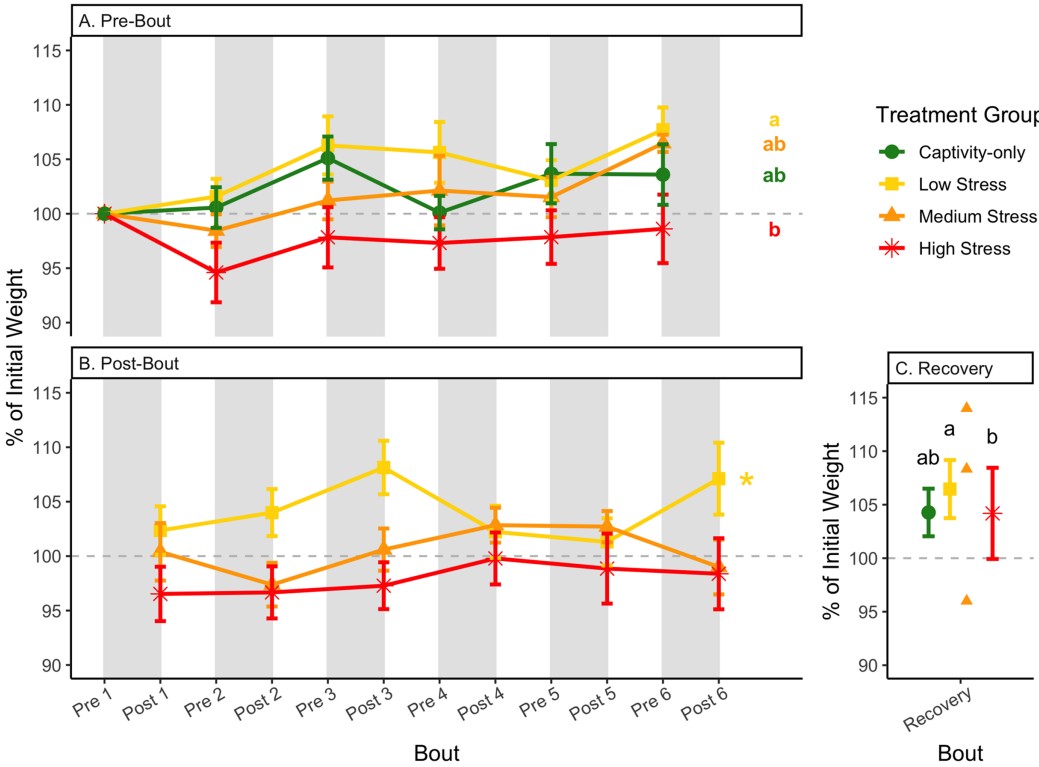

**Figure 4 Weight changes during 6 months of chronic stress and after 6 weeks of recovery.** Birds were subjected to six bouts of chronic stress of varying length (depicted as grey regions in (A) and (B)) and intensity, depending on treatment group. Weights were measured at the start (A) and end (B) of each bout, as well as after 6 weeks of chronic stress (C). Graphed values represent percent change from initial weight, but statistics were performed on raw values. In (A) and (B), asterisks indicate treatment groups with a significant effect ($p < 0.05$) of bout and letters indicate *post hoc* analysis across groups. Annotation in (C) reflects significant differences ($p < 0.05$) from an ANOVA. The medium stress group was excluded from recovery analysis due to low sample size but is graphed for completeness.

# RESULTS

## All data from 6 months of chronic stress

Results from the overall statistical model are shown in Table 1 (columns 3–9), in which each metric (weight plus fed, fasted, and fold-change measurements for all three metabolites) were compared against treatment group, pre/post-bout, bout number, and every interaction. Every metric was significantly affected by treatment group (Table 1, column 3) or an interaction with treatment group (Table 1, columns 6, 7, 9), indicating that the intensity and duration of stress affected both weight and metabolites (Figs. 4–13A and Figs. 4–13B, comparing different treatment lines). Similarly, every metric, except for fold-change in uric acid, was significantly affected by a sample being taken at the beginning or end of a bout of chronic stress (Table 1, column 4) or an interaction (Table 1, columns 6, 8, 9), indicating changes during each bout of stress that recovered during each break (Figs. 4–13, comparing panel A to panel B). Lastly, every metric except for fold-change in glucose and fed uric acid significantly changed with bout number (Table 1, column 5) or an

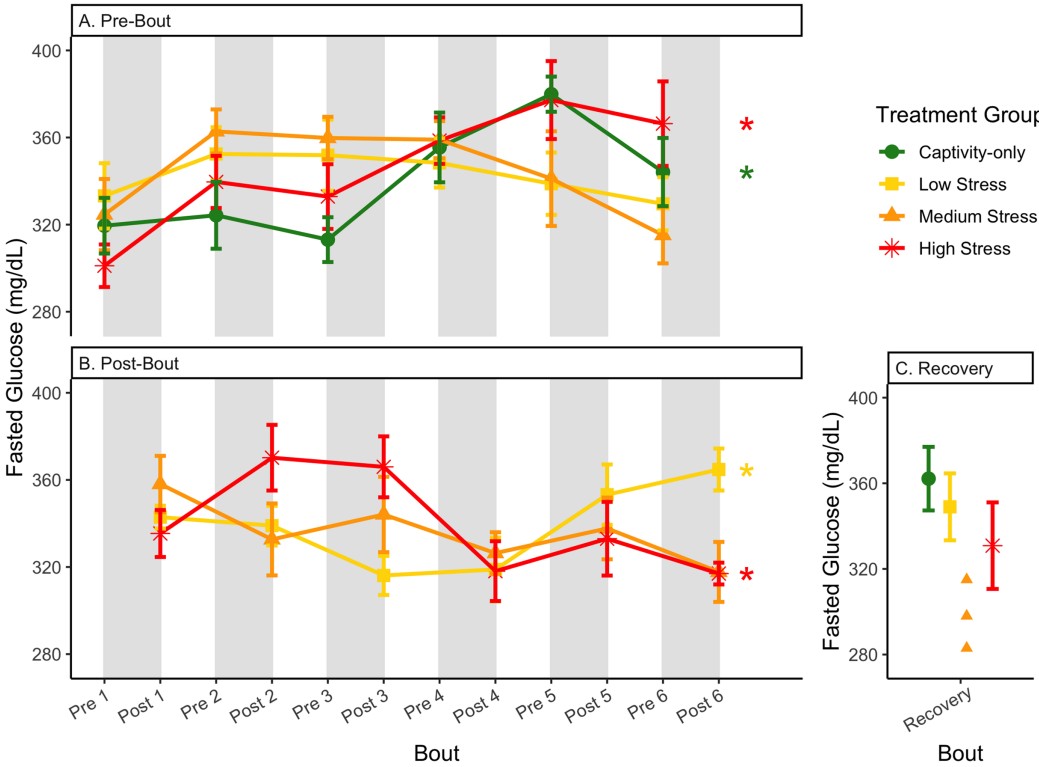

**Figure 5 Changes in blood glucose in a fasted state through 6 months of chronic stress and after 6 weeks of recovery.** Birds were subjected to six bouts of chronic stress of varying length (depicted as grey regions in (A) and (B)) and intensity, depending on treatment group. Blood samples for metabolite analysis were measured at the start (A) and end (B) of each bout, as well as after 6 weeks of chronic stress (C). In (A) and (B), asterisks indicate treatment groups with a significant effect ($p < 0.05$) of bout. The medium stress group was excluded from recovery analysis due to low sample size but is graphed for completeness.

interaction with bout number (Table 1, columns 7, 8, 9), indicating changes as the long-term chronic stress progressed (Figs. 4–13, comparing across the x-axes).

Because fold-change uric acid (Fig. 13) was not affected by pre/post-bout, we removed that factor from the model and re-ran it. The results of the new model, "fold-change uric acid~group * bout num", indicate that treatment group, bout number, and the interaction all affect fold-change uric acid (Table 2, columns 3–5; Figs. 13A and 13B, comparing across treatment lines and across the x-axis). However, when the groups are separated, only the captivity-only and the medium stress group show a significant effect of bout number (Table 2, columns 6–9).

## Pre-bout data (Figs. 4–13A)

In samples taken at the beginning of each bout, there were differences in treatment groups (or an interaction with treatment group) for every metric except for fed glucose, fasted ketones, and fed uric acid (Table 3, columns 3, 5; Figs. 4–13A, comparing different treatment lines). No obvious patterns arose among which treatment groups significantly changed with bout number (Table 3, columns 6–8, Figs. 4–13A, comparing each line across the x-axis).

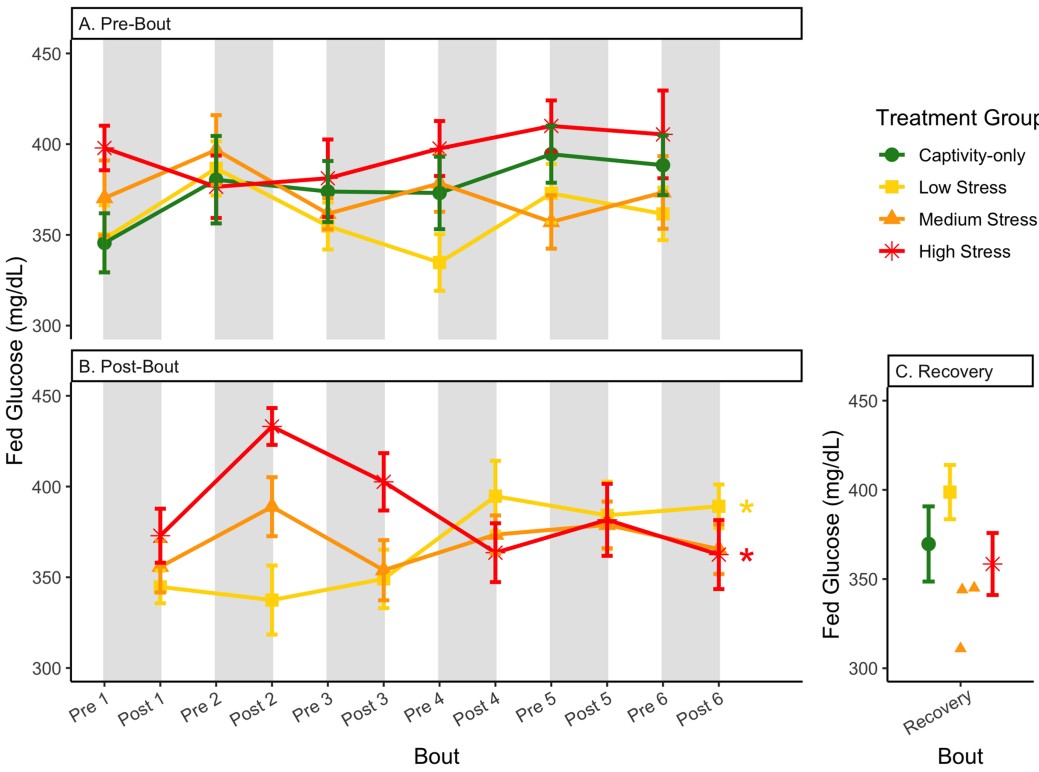

**Figure 6 Changes in blood glucose in a fed state through 6 months of chronic stress and after 6 weeks of recovery.** Birds were subjected to six bouts of chronic stress of varying length (depicted as grey regions in (A) and (B)) and intensity, depending on treatment group. Blood samples for metabolite analysis were measured at the start (A) and end (B) of each bout, as well as after 6 weeks of chronic stress (C). In (B), asterisks indicate treatment groups with a significant effect ($p < 0.05$) of bout. The medium stress group was excluded from recovery analysis due to low sample size but is graphed for completeness.

## Post-bout data (Figs. 4–13B)

In samples taken at the end of each bout, every metric showed a significant interaction between group and bout number (Table 4, column 5; Figs. 4–13A, comparing lines), except for fold-change glucose, which was only significantly affected by bout number (Table 4, column 4). The low stress group exhibited a significant effect of bout number for weight (Fig. 4B, yellow line), fasted glucose (Fig. 5B, yellow line), and fed glucose (Fig. 6B, yellow line), while the medium and high stress groups exhibited significant effects of bout in ketones and uric acid (Table 4, columns 6–8; Figs. 7–12B, orange and yellow lines).

## Recovery

After 6 weeks of recovery, there was still a difference in weights among groups (Table 1, column 10; Fig. 4C), however no differences persisted among any measures of metabolites (Table 1, column 10; Figs. 4–13C). The eta$^2$ effect size for weights at the recovery timepoint was 0.31.

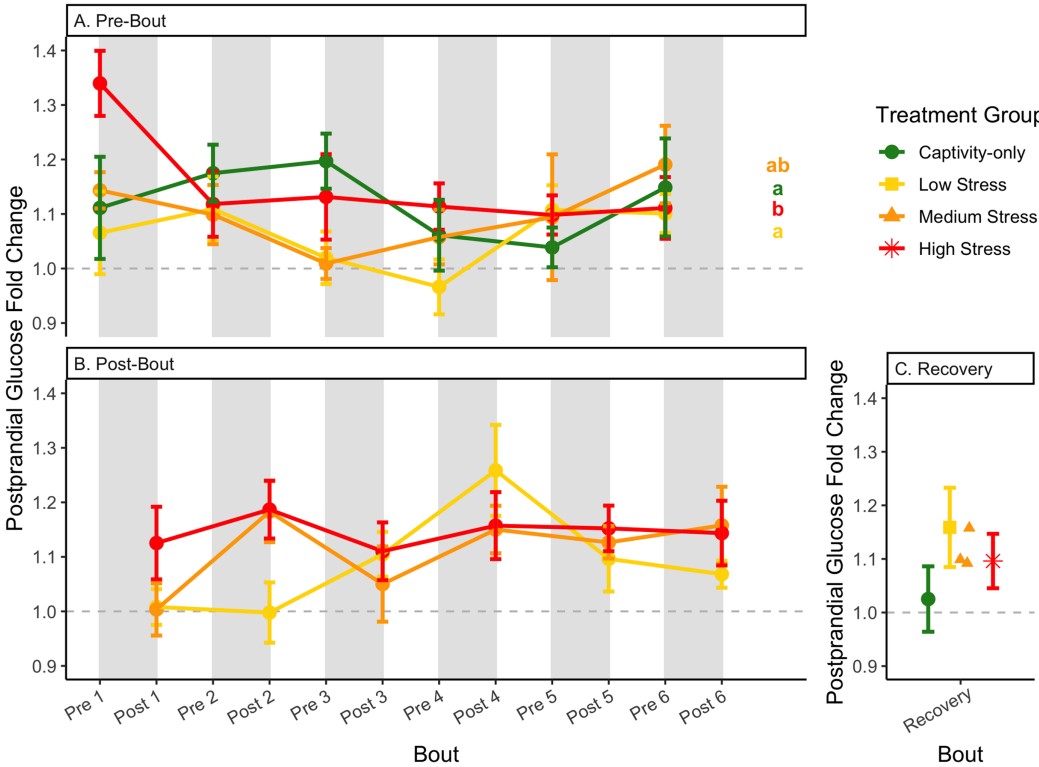

**Figure 7 Changes in postprandial fold-change blood glucose through 6 months of chronic stress and after 6 weeks of recovery.** Fold-change was calculated by dividing fed blood glucose by fasted blood glucose. Birds were subjected to six bouts of chronic stress of varying length (depicted as grey regions in (A) and (B)) and intensity, depending on treatment group. Blood samples for metabolite analysis were measured at the start (A) and end (B) of each bout, as well as after 6 weeks of chronic stress (C). In (A), letters indicate *post hoc* analysis across groups. The medium stress group was excluded from recovery analysis due to low sample size but is graphed for completeness.

## DISCUSSION

### Group differences

The purpose of this study was to determine if differing intensities of chronic stress produced different effects on metabolism and whether these changes could be used to infer the timing of a transition from reactive homeostasis to homeostatic overload. Consequently, the critical comparisons were the main effects or interactions with treatment groups: low, medium, or high chronic stress. Overall, weight and the metabolic effects of long-term chronic stress were both dependent on the intensity of the stress, but the effects were not consistent (summarized in Table 5). This study was part of a larger project that assessed differing intensities of chronic stress on a variety of chronic stress-related metrics. For example, the high stress treatment, but not medium, low, or captivity-control treatments, had a significant impact on immune function (*Beattie et al., 2023*).

There are few studies on stressor intensity and energy usage and those that exist focus on acute stress. In laboratory rats, graded intensities of acute stressors produce concurrent

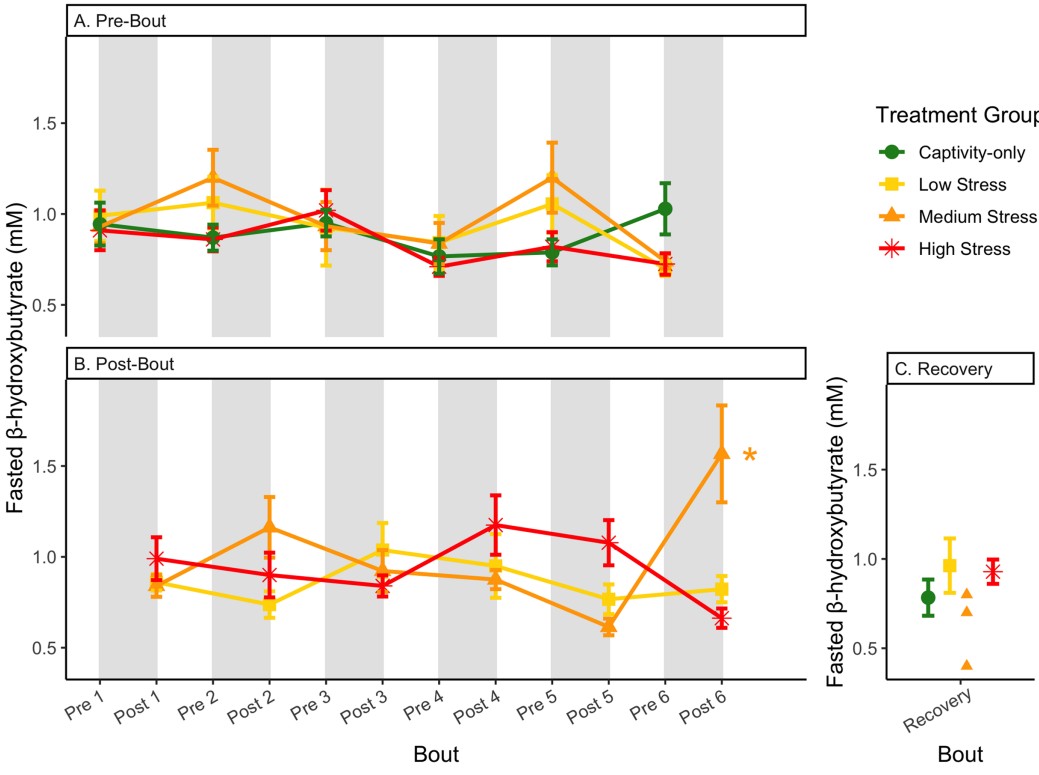

**Figure 8 Changes in blood ketones (β-hydroxybutyrate) in a fasted state through 6 months of chronic stress and after 6 weeks of recovery.** Birds were subjected to six bouts of chronic stress of varying length (depicted as grey regions in (A) and (B)) and intensity, depending on treatment group. Blood samples for metabolite analysis were measured at the start (A) and end (B) of each bout, as well as after 6 weeks of chronic stress (C). In (B), asterisks indicate treatment groups with a significant effect ($p < 0.05$) of bout. The medium stress group was excluded from recovery analysis due to low sample size but is graphed for completeness.

graded responses of food consumption (*Martí, Martí & Armario, 1994*), glucose (*Armario, Montero & Balasch, 1986*; but not sometimes: *Márquez, Belda & Armario, 2002*) and free fatty acids, but not of triglycerides (*Armario, Montero & Balasch, 1986*). Captive European starlings show longer heart rate recovery times following the conclusion of stronger acute stressors (*Nephew, Kahn & Romero, 2003*), and since heart rate correlates with energy consumption in this species (*Cyr, Wikelski & Romero, 2008*), stronger acute stressors appear to require more acute energy expenditure. Existing data are less consistent for chronic stressors. Graded responses to chronic stress persisted with food intake (*Martí, Martí & Armario, 1994*) but not glucose (*Armario, Marti & Gil, 1990*) in rats. However, these two studies are not perfectly comparable to ours because they purposefully administered the same stressor, at the same time, every day for 27 days, whereas our stressors and stressor times were randomized to minimize the potential for habituation. Given the glucose data, *Armario, Marti & Gil (1990)* concluded that rats habituate to this protocol of chronic stress, but the food intake data (*Martí, Martí & Armario, 1994*), argue that they do not. Data from the present study, therefore, removes the confounding variable

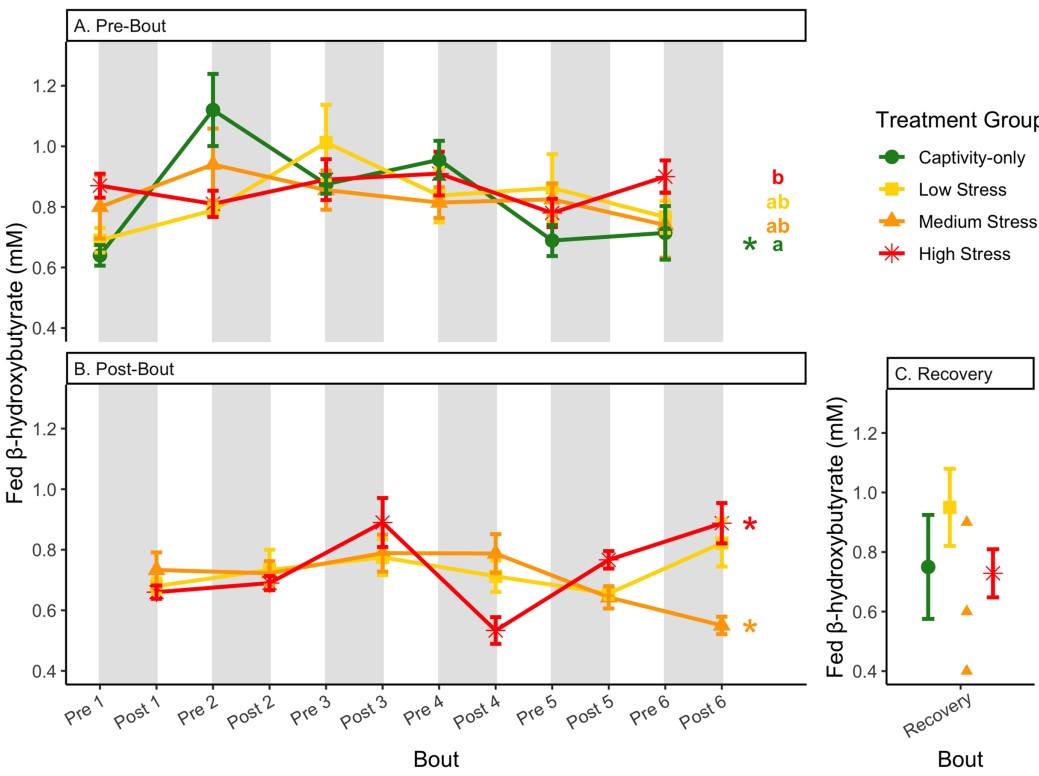

**Figure 9 Changes in blood ketones (β-hydroxybutyrate) in a fed state through 6 months of chronic stress and after 6 weeks of recovery.** Birds were subjected to six bouts of chronic stress of varying length (depicted as grey regions in (A) and (B)) and intensity, depending on treatment group. Blood samples for metabolite analysis were measured at the start (A) and end (B) of each bout, as well as after 6 weeks of chronic stress (C). In (A) and (B), asterisks indicate treatment groups with a significant effect ($p < 0.05$) of bout and letters indicate *post hoc* analysis across groups. The medium stress group was excluded from recovery analysis due to low sample size but is graphed for completeness.

of habituation and suggests that graded responses to chronic stress do exist, though they are not consistent across metrics.

## Long-term chronic stress

One hallmark of chronic stress is a decrease in body weight, which can be dependent on the type of chronic stress protocol (summarized in *Dickens & Romero, 2013*). Chronic stress protocols using foot-shock or variable stressors (such as used here) almost always lead to a decrease in body weight (though not in this study), whereas chronic restraint stress and chronic social stress have more variable results (*Dickens & Romero, 2013*). Though it is difficult to know specifically how animals perceive stressors, the different effects on body weight could be interpreted as being caused by differing intensities of the stressors. In addition, once a period of chronic stress ends, body weight is overcompensated, such that it is higher than the pre-stress weight (*Awerman & Romero, 2010*; *Cyr & Romero, 2007*; *Rich & Romero, 2005*). In the present study, a significant difference between pre-bout and post-bout weights (Table 1, columns 4, 6, 8, 9; Figs. 4A and 4B) indicates that within each bout of stress, there was a change in weight (generally a

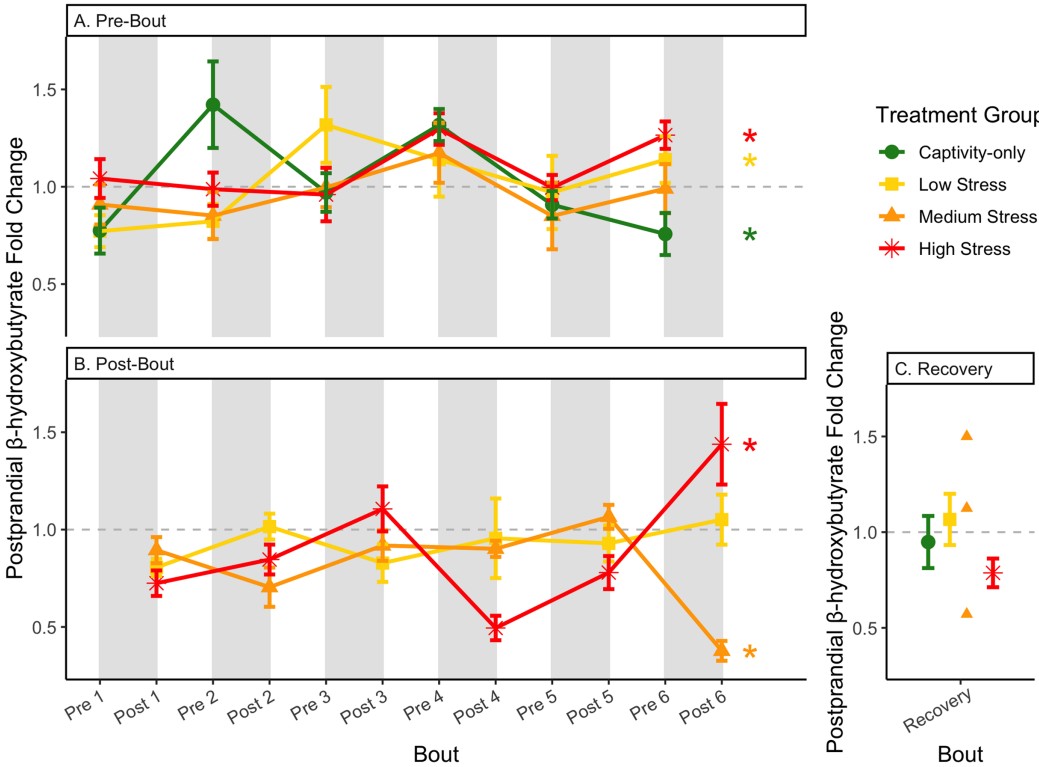

**Figure 10 Changes in postprandial fold-change blood ketones (b-hydroxybutyrate) through 6 months of chronic stress and after 6 weeks of recovery.** Fold-change was calculated by dividing fed blood β-hydroxybutyrate by fasted blood β-hydroxybutyrate. Birds were subjected to six bouts of chronic stress of varying length (depicted as grey regions in (A) and (B)) and intensity, depending on treatment group. Blood samples for metabolite analysis were measured at the start (A) and end (B) of each bout, as well as after 6 weeks of chronic stress (C). In (A) and (B), asterisks indicate treatment groups with a significant effect ($p < 0.05$) of bout. The medium stress group was excluded from recovery analysis due to low sample size but is graphed for completeness.

decrease), though it does not seem to be as dramatic as the Awerman and Romero study (2010).

In an earlier study in house sparrows, a single bout of chronic stress resulted in an increase in plasma glucose and ketones, in a fasted state, and plasma glucose and uric acid in a fed state (*Beattie et al., 2022*). Interestingly, these changes did not return to pre-stress levels when measured up to 2 weeks after the chronic stress ended. Other studies (*Awerman & Romero, 2010*; *Cyr et al., 2007*; *Fokidis et al., 2011*; *Kern et al., 2007*; *Xie et al., 2015*) showed no change in plasma glucose with chronic stress. We were unable to find any other papers on changes in ketones with chronic stress in birds, but changes in fat scores suggest that fat mobilization can be affected by chronic stress (*Boyer & MacDougall-Shackleton, 2020*). Uric acid has been reported to not change (*Xie et al., 2015*), increase (*Gormally et al., 2019a*), or decrease (*Gormally et al., 2019b, 2018*). Uric acid has also been shown to not change with one bout of chronic stress but then decrease upon a second bout of chronic stress (*Awerman & Romero, 2010*). The consensus in the literature is that glucose, ketones, and uric acid do not change in a predictable manner with chronic stress and our data corroborate that conclusion.

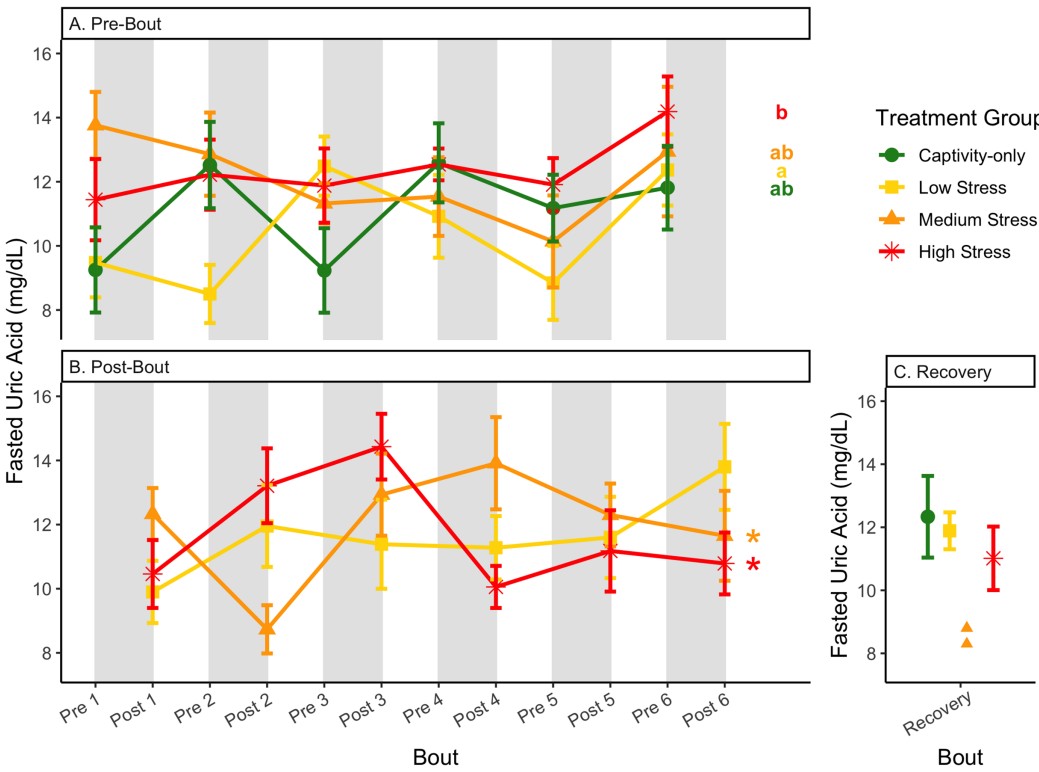

**Figure 11 Changes in uric acid in a fasted state through 6 months of chronic stress and after 6 weeks of recovery.** Birds were subjected to six bouts of chronic stress of varying length (depicted as grey regions in (A) and (B)) and intensity, depending on treatment group. Blood samples for metabolite analysis were measured at the start (A) and end (B) of each bout, as well as after 6 weeks of chronic stress (C). In (A) and (B), asterisks indicate treatment groups with a significant effect ($p < 0.05$) of bout and letters indicate *post hoc* analysis across groups. The medium stress group was excluded from recovery analysis due to low sample size but is graphed for completeness.

In the present study, all three metabolites showed effects of bout number (Table 3; column 5, 7–9) and pre/post bout (samples taken at the start or end of the bout; Table 3, columns 4, 6, 8, 9), but the direction of these changes were inconsistent (Table 5). Importantly, our study took place over 6 months, whereas the *Awerman & Romero (2010)* study was 3 months and most other chronic stress studies are less than 1 month long. The longer length and repeated stress periods may explain the inconsistencies over time. While it is possible that the extended length of our study exposed many subtle differences, it is unclear whether these changes are biologically relevant. Despite many significant *p*-values in the present study, the main conclusion is that there is no consistent response that is stratified by stressor intensity.

### Recovery at the end of six bouts of chronic stress

The only metric in which group differences persisted 6 weeks after the experiment ended was weight (Fig. 4C; Table 1, column 10). Any differences in metabolites across treatment groups did not persist. The high stress group added some weight during the recovery period, but not enough to match the low stress group. In other words, the intensity of long-term stress did not affect long-long term recovery of metabolites, but it did for weight.

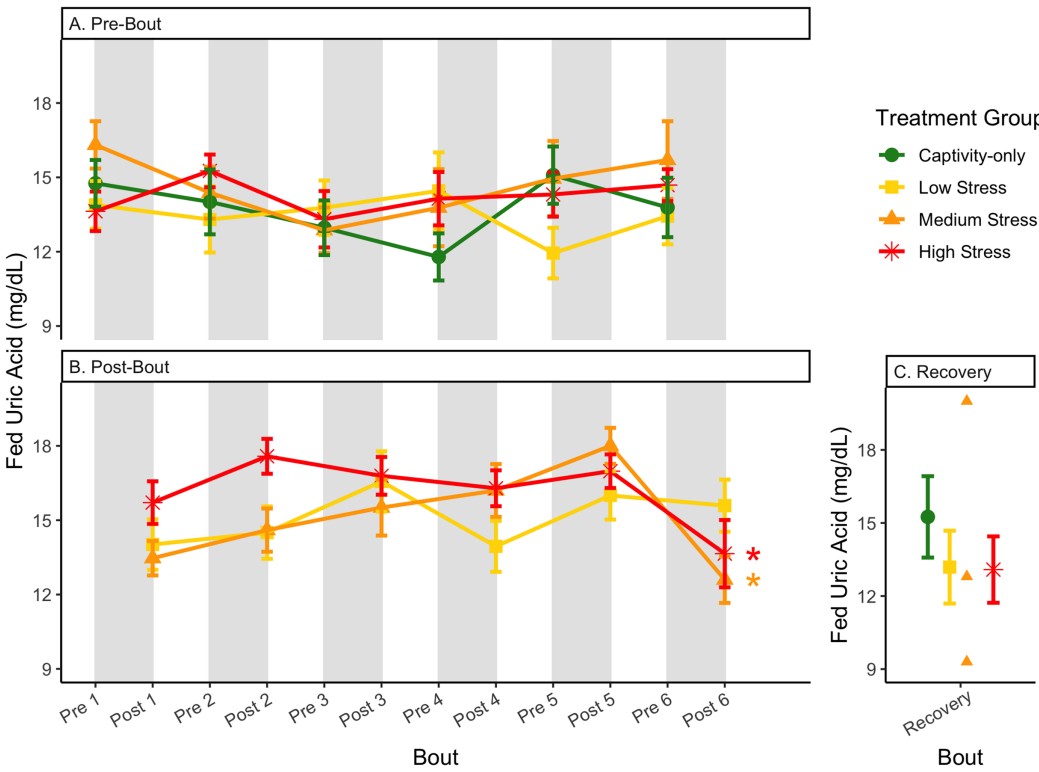

**Figure 12 Changes in blood uric acid in a fed state through 6 months of chronic stress and after 6 weeks of recovery.** Birds were subjected to six bouts of chronic stress of varying length (depicted as grey regions in (A) and (B)) and intensity, depending on treatment group. Blood samples for metabolite analysis were measured at the start (A) and end (B) of each bout, as well as after 6 weeks of chronic stress (C). In (B), asterisks indicate treatment groups with a significant effect ($p < 0.05$) of bout. The medium stress group was excluded from recovery analysis due to low sample size but is graphed for completeness.

The partial, but deficient, weight gain with high stress suggests that weight is not completely driven by changes in metabolism during the recovery from long-term chronic stress. This adds to the recent finding that changes in weight with corticosterone exposure are not mediated by metabolism (*Bauer et al., 2022*). Although glucose and heart rate during recovery from an acute stressor reflect the intensity of that stressor (*Márquez, Belda & Armario, 2002*; *Nephew, Kahn & Romero, 2003*), to our knowledge work on the recovery from differing intensities of chronic stress has not been performed previously. Specifically for glucose, the lack of differences across treatment groups may reflect that glucose is very tightly regulated in birds (*Alonso-Alvarez & Ferrer, 2001*; *Castellini & Rea, 1992*; *Rodríguez, Tortosa & Villafuerte, 2005*) and thus, any changes that a prolonged chronic stress event produces are quickly reversed. This may be especially true if the animal did not enter homeostatic overload during the chronic stress period.

## Feeding effects

To investigate changes after feeding, we calculated the fold-change of each fasted sample to its respective fed sample. We would expect an increase in plasma glucose (Fig. 2A) (*Scanes, 2015b*) and uric acid (from food-derived uric acid; Fig. 2C) and a decrease in ketones

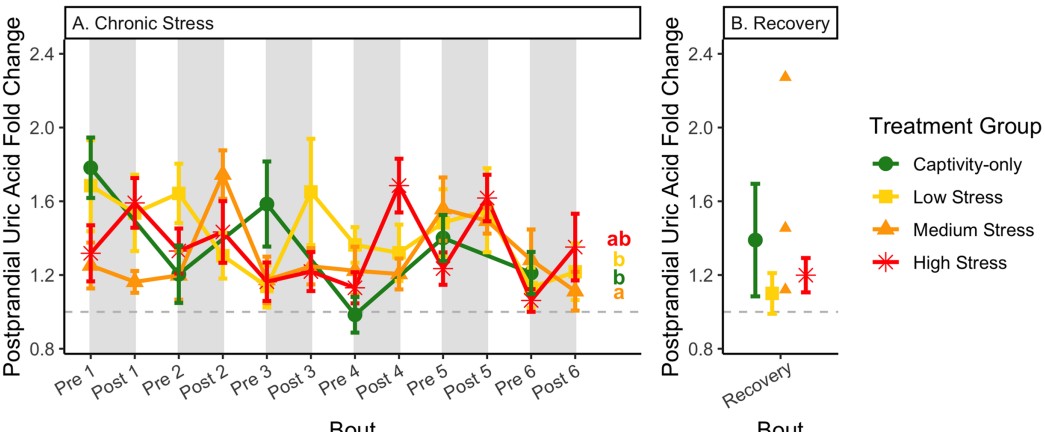

**Figure 13 Changes in postprandial fold-change blood uric acid through 6 months of chronic stress and after 6 weeks of recovery.** Fold-change was calculated by dividing fed blood uric acid by fasted blood uric acid. Birds were subjected to six bouts of chronic stress of varying length (depicted as grey regions in (A)) and intensity, depending on treatment group. Blood samples for metabolite analysis were measured at the start and end of each bout (A), as well as after 6 weeks of chronic stress (B). There was no significant difference between samples taken at the start *vs.* end of the bout, so they are graphed together. In (A), letters indicate *post hoc* analysis across groups ($p < 0.05$). The medium stress group was excluded from recovery analysis due to low sample size but is graphed for completeness.

**Table 2 Results of secondary analysis for fold-change uric acid.**

| (1) Metric | (2) | (3) Group | (4) BoutNum | (5) Group * BoutNum | (6) Captivity-only | (7) Low stress | (8) Medium stress | (9) High stress |
|---|---|---|---|---|---|---|---|---|
| Uric acid | Fold-change | $F_{3,324} = 4.32$ | $F_{5,319} = 3.82$ | $F_{15,322} = 1.59$ | $F_{5,40} = 3.48$ | $F_{5,95} = 1.44$ | $F_{5,85} = 2.66$ | $F_{5,99} = 1.36$ |
| | | $p = 2.25 \times 10^{-3}$ | $p = 5.26 \times 10^{-3}$ | $p = 0.07$ | $p = 0.01$ | $p = 0.22$ | $p = 0.03$ | $p = 0.24$ |

**Note:**
Because there was no effect of pre/post-bout on fold-change uric acid in the main model (shown in Table 1), pre/post-bout was removed from the model. The results of a new model, "fold-change uric acid~group * bout num" with bird identity as a random effect, are shown in columns 3–5. Because there was a significant effect of group (and a significant interaction with group), the four treatment groups were split and four independent models were run. These four models were "fold-change uric acid~bout num" with bird identity as a random effect, and the results are down in columns 6–9. Cells that are bolded contain *p*-values less than 0.05.

(signaling the end of fasting; Fig. 2B) (*Buyse & Decuypere, 2015*) after feeding (*Beattie et al., 2022*; *Romero & Wingfield, 2016*).

The postprandial rise in glucose differed across groups and depended on pre/post-bout (samples taken at the start or end of the bout; Fig. 7; Table 1, columns 3, 4). In general, each bout of chronic stress increased the postprandial rise in glucose (Table 4, Column 4; Fig. 7B) but was not maintained in the following pre-bout samples (no effect of bout number Table 3, column 4; Fig. 7A). This indicates that the larger postprandial rise at the end of the chronic stress bout quickly recovered by the onset of the next bout. This further supports the idea that, although bouts of chronic stress might cause changes in blood glucose (*Beattie et al., 2022*), the tight regulation of glucose means that recovery to normal levels happens quickly (*Alonso-Alvarez & Ferrer, 2001*; *Castellini & Rea, 1992*; *Rodríguez, Tortosa & Villafuerte, 2005*).

Conversely, the postprandial shutdown of endogenous fat breakdown (as measured by β-hydroxybutyrate) showed significant changes across groups (though they were

**Table 3 Results from data analyses on samples taken at the beginning of each bout (pre-bout).**

| (1) Metric | (2) | (3) Group | (4) BoutNum | (5) Group * BoutNum | (6) Captivity-only | (7) Low stress | (8) Medium stress | (9) High stress |
|---|---|---|---|---|---|---|---|---|
| Weight | | $F_{3,117}=5.90$, $p=8.70\times10^{-4}$ | $F_{5,157}=2.18$, $p=0.06$ | $F_{15,157}=1.58$, $p=0.08$ | | | | |
| Glucose | Fasted | $F_{2,172}=0.86$, $p=0.46$ | $F_{5,158}=4.05$, $p=1.75\times10^{-3}$ | $F_{15,158}=1.99$, $p=0.02$ | $F_{5,40}=2.69$, $p=7.66\times10^{-3}$ | $F_{5,40}=0.77$, $p=0.57$ | $F_{5,37}=1.81$, $p=0.14$ | $F_{5,41}=4.76$, $p=1.58\times10^{-3}$ |
| | Fed | $F_{3,162}=2.15$, $p=0.10$ | $F_{5,158}=1.41$, $p=0.22$ | $F_{15,159}=0.98$, $p=0.48$ | | | | |
| | Fold-change | $F_{3,189}=3.83$, $p=0.01$ | $F_{5,159}=1.11$, $p=0.35$ | $F_{15,159}=1.14$, $p=0.33$ | | | | |
| Ketones | Fasted | $F_{3,141}=0.10$, $p=0.96$ | $F_{5,155}=1.03$, $p=0.40$ | $F_{15,154}=1.62$, $p=0.07$ | | | | |
| | Fed | $F_{3,179}=3.06$, $p=0.03$ | $F_{5,157}=7.89$, $p=1.19\times10^{-6}$ | $F_{15,157}=2.21$, $p=7.88\times10^{-3}$ | $F_{5,39}=7.01$, $p=9.22\times10^{-5}$ | $F_{5,38}=2.32$, $p=0.06$ | $F_{5,36}=0.53$, $p=0.75$ | $F_{5,43}=0.98$, $p=0.44$ |
| | Fold-change | $F_{3,176}=1.64$, $p=0.18$ | $F_{5,155}=4.88$, $p=3.57\times10^{-4}$ | $F_{15,155}=2.04$, $p=0.02$ | $F_{5,38}=5.15$, $p=1.04\times10^{-3}$ | $F_{5,36}=3.08$, $p=0.02$ | $F_{5,36}=0.84$, $p=0.53$ | $F_{5,43}=2.87$, $p=0.03$ |
| Uric acid | Fasted | $F_{3,174}=3.66$, $p=0.01$ | $F_{5,160}=2.11$, $p=0.07$ | $F_{15,160}=1.54$, $p=0.10$ | | | | |
| | Fed | $F_{3,123}=1.32$, $p=0.27$ | $F_{5,159}=1.82$, $p=0.11$ | $F_{15,159}=1.23$, $p=0.25$ | | | | |
| | Fold-change | | | | | | | |

Note:
Columns 3–5 show the results of the linear mixed-effect model, "metric~group * bout num", with bird identity as a random effect. Metrics are listed in columns 1–2, group is treatment group (captivity-only, low stress, medium stress, high stress), and bout num is the bout number (1–6). There are no results for fold-change uric acid because there was no significant effect of group or significant interaction with group in the main analysis (shown in Table 1). For the model, "metric~group * bout num", if there was a significant effect of group or a significant interaction with group, the groups were split and four separate models (one for each group) was run, in which the model was "metric~bout num", with bird identity as a random effect. The results from these four models are shown in columns 6–9. There are no results in columns 6–9 for fed glucose, fasted ketones, and fed uric acid because the previous model ("metric~group * bout num") did not show a significant effect of group or a significant interaction with group. Cells that are bolded contain $p$-values less than 0.05.

**Table 4 Results from data analyses on samples taken at the end of each bout (post-bout).**

| (1) Metric | (2) | (3) Group | (4) BoutNum | (5) Group * BoutNum | (6) Low stress | (7) Medium stress | (8) High stress |
|---|---|---|---|---|---|---|---|
| Weight | | $F_{2,73}=2.20$, $p=0.12$ | $F_{5,110}=4.83$, $p=4.93\times10^{-4}$ | $F_{10,111}=2.97$, $p=2.39\times10^{-3}$ | $F_{5,39}=4.67$, $p=1.98\times10^{-3}$ | $F_{5,31}=1.76$, $p=0.15$ | $F_{5,42}=1.14$, $p=0.35$ |
| Glucose | Fasted | $F_{2,136}=0.91$, $p=0.41$ | $F_{5,115}=2.29$, $p=0.05$ | $F_{10,116}=2.51$, $p=9.05\times10^{-3}$ | $F_{5,40}=3.55$, $p=9.44\times10^{-3}$ | $F_{5,34}=0.85$, $p=0.52$ | $F_{5,42}=1.99$, $p=0.02$ |
| | Fed | $F_{2,124}=0.94$, $p=0.39$ | $F_{5,113}=3.33$, $p=7.61\times10^{-3}$ | $F_{10,114}=3.30$, $p=8.69\times10^{-4}$ | $F_{5,40}=3.56$, $p=9.36\times10^{-3}$ | $F_{5,33}=1.15$, $p=0.35$ | $F_{5,42}=3.33$, $p=0.01$ |
| | Fold-change | $F_{2,137}=1.79$, $p=0.17$ | $F_{5,113}=3.16$, $p=0.01$ | $F_{10,116}=1.30$, $p=0.24$ | | | |
| Ketones | Fasted | $F_{2,130}=0.66$, $p=0.52$ | $F_{5,111}=1.07$, $p=0.38$ | $F_{10,112}=3.94$, $p=1.28\times10^{-4}$ | $F_{5,38}=1.22$, $p=0.32$ | $F_{5,32}=6.12$, $p=4.50\times10^{-4}$ | $F_{5,41}=2.35$, $p=0.06$ |
| | Fed | $F_{2,126}=0.59$, $p=0.56$ | $F_{5,113}=1.63$, $p=0.16$ | $F_{10,114}=4.09$, $p=7.91\times10^{-5}$ | $F_{5,39}=1.64$, $p=0.17$ | $F_{5,33}=3.31$, $p=0.02$ | $F_{5,42}=7.70$, $p=3.27\times10^{-5}$ |

(Continued)

| (1) Metric | (2) | (3) Group | (4) BoutNum | (5) Group * BoutNum | (6) Low stress | (7) Medium stress | (8) High stress |
|---|---|---|---|---|---|---|---|
| | Fold-change | $F_{2,132} = 0.79$ $p = 0.46$ | $F_{5,110} = 1.00$ $p = 0.42$ | $F_{10,112} = 5.36$ $p = 1.91 \times 10^{-6}$ | $F_{5,38} = 0.86$ $p = 0.51$ | $F_{5,32} = 5.62$ $p = 8.07 \times 10^{-4}$ | $F_{5,41} = 8.46$ $p = 1.41 \times 10^{-5}$ |
| Uric acid | Fasted | $F_{2,105} = 1.47$ $p = 0.23$ | $F_{5,114} = 1.81$ $p = 0.11$ | $F_{10,115} = 3.45$ $p = 5.43 \times 10^{-4}$ | $F_{5,40} = 1.54$ $p = 0.20$ | $F_{5,33} = 5.05$ $p = 1.51 \times 10^{-3}$ | $F_{5,42} = 3.40$ $p = 0.01$ |
| | Fed | $F_{2,115} = 1.47$ $p = 0.23$ | $F_{5,113} = 1.63$ $p = 0.16$ | $F_{10,115} = 2.05$ $p = 0.03$ | $F_{5,40} = 1.39$ $p = 0.25$ | $F_{5,32} = 4.31$ $p = 4.02 \times 10$ | $F_{5,42} = 2.63$ $p = 0.04$ |
| | Fold-change | | | | | | |

**Note:**
Columns 3–5 show the results of the linear mixed-effect model, "metric~group * bout num", with bird identity as a random effect. Metrics are listed in columns 1–2, group is treatment group (captivity-only, low stress, medium stress, high stress), and bout num is the bout number (1–6). There are no results for fold-change uric acid because there was no significant effect of group or significant interaction with group in the main analysis (shown in Table 1). For the model, "metric~group * bout num", if there was a significant effect of group or a significant interaction with group, the groups were split and four separate models (one for each group) was run, in which the model was "metric~bout num", with bird identity as a random effect. The results from these four models are shown in columns 6–8. There are no results in columns 6–9 for fold-change glucose because the previous model ("metric~group * bout num") did not show a significant effect of group or a significant interaction with group. There is no column for the captivity-only group because they were only sampled at pre-bout timepoints to reduce disturbance. Cells that are bolded contain p-values less than 0.05.

**Table 5 Summary of results.**

| (1) Metric | (2) | (3) Pre/post-bout | (4) Group (pre-bout) | (5) Group (post-bout) | (6) Bout (pre-bout) | (7) Bout (post-bout) |
|---|---|---|---|---|---|---|
| Weight | | ↑ ↓ – | ↓ | ↓ | – | ↑ – |
| Glucose | Fasted | ↑ ↓ – | – | ↓ | ↑ – | ↑ ↓ – |
| | Fed | ↑ ↓ – | – | ↓ | – | ↑ ↓ – |
| | Fold-change | ↑ ↓ – | – | – | – | ↑ – |
| Ketones | Fasted | ↑ – | – | – | – | ↑ – |
| | Fed | ↓ – | – | – | ↓ – | ↓ – |
| | Fold-change | ↓ – | – | – | ↑ ↓ – | ↑ ↓ – |
| Uric acid | Fasted | ↑ ↓ – | – | – | – | ↑ ↓ – |
| | Fed | ↑ ↓ – | – | – | – | ↓ – |
| | Fold-change | – | – | – | ↓ – | |

**Note:**
Column 3 reflects direction of changes from pre-bout samples to post-bout samples. Columns 4 and 5 reflect direction of changes with increasing chronic stress intensity in both the pre-bout subset and the post-bout subset. Columns 6 and 7 reflect direction of changes throughout time (bout number) in both the pre-bout subset and the post-bout subset. ↑ means an increase, ↓ means a decrease, – means no change or no clear pattern, ↑↓ means an increase for some bouts, but a decrease in others. Note that fold-change uric acid did not show a significant effect of pre/post-bout, so the changes with group (col. 4 and 5) and bout (col. 6 and 7) were determined with the pre- and post-bout data combined.

inconsistent with increasing stress intensity; Table 1, columns 6, 7, 9; Fig. 10), pre/post-bout (Table 1, columns 4, 6, 8, 9; Fig. 10), and throughout the long-term stress experiment (showing both increases and decreases; Table 5). House sparrows enter phase II of fasting overnight (*Beattie et al., 2022*; *Cherel et al., 1988*; *Khalilieh, McCue & Pinshow, 2012*; *Romero & Wingfield, 2016*) to fuel basic life processes through the breakdown of fat stores. As soon as refeeding occurs, the animal should switch energy sources from fat stores to the just-consumed food, meaning that plasma ketones should decrease (and thus, the postprandial fold-change should be less than one). Interestingly, most of the data was less

than one in post-bout samples (Fig. 10B), indicating that after a bout of stress, the birds adequately shut down their fat breakdown after a meal. The pre-bout data was around one or higher than one (Fig. 10A), indicating that after a break from stress, there is a latent effect in the postprandial shutdown of fat metabolism. This could be an example of homeostatic overload as the prolonged fat breakdown, when it is not necessary, would be indicative of pathology (*Romero, Dickens & Cyr, 2009*). This effect was not seen with a single bout of chronic stress (*Beattie et al., 2022*) and so appears to be a result of the long-term chronic stress of the present protocol.

Although the birds in this study were provided a mixed diet of sunflower seeds and millet, they preferentially eat the sunflower seeds. Because sunflower seeds are very high in uric acid (*Hafez, Abdel-Rahman & Naguib, 2017*), we used fed uric acid levels as a proxy of food consumption (*Beattie et al., 2022*). In contrast, when birds are not consuming sunflower seeds, *i.e.*, when fasting, uric acid is an indicator of endogenous protein breakdown. There was no significant difference in fold-change uric acid between samples taken at the start of a stress bout and at the end of a stress bout (Table 1, columns 4, 6, 8, 9; Fig. 13A), meaning that any changes (sometimes increases, sometimes decreases) in food consumption during bouts of stress did not recover within the short break. This is consistent with a study that showed that fed uric acid did not recover 2 weeks after a single bout of chronic stress ended (*Beattie et al., 2022*).

## CONCLUSIONS

The reactive scope model predicts that increasing intensities of chronic stress will result in increasing degrees of wear-and-tear, resulting in different timing of entering homeostatic overload. Despite many significant effects of pre/post-bout samples, groups, and bout number, there were no clear overall trends in the data (Table 5). In other words, the data reflected an unpredictable dysregulation of metabolism, but there were no metabolic changes that consistently correlated with the stratification of chronic stress intensity over time. Consequently, the data did not support our hypothesis that increasing stress intensity would cause increased chronic stress symptoms. This conclusion suggests that metabolism may, in fact, not be sensitive to the intensity of chronic stress and should not be used as a proxy measurement.

## ACKNOWLEDGEMENTS

We thank Emma Rosen, Stephen Kennedy, Andrew Kung, Bradley Pedro, Rachel Riccio, Ruby Guo, and Denyelle Kilgour for helping apply stressors and take blood samples. We also thank the Tufts University animal care staff.

### Funding

Funding was provided by National Science Foundation grant IOS-1655269 to L. Michael Romero. The funders had no role in study design, data collection and analysis, decision to publish, or preparation of the manuscript.

## Grant Disclosures

The following grant information was disclosed by the authors:
Funding provided by National Science Foundation: IOS-1655269.

## Competing Interests

The authors declare that they have no competing interests.

## Author Contributions

- Ursula K. Beattie conceived and designed the experiments, performed the experiments, analyzed the data, prepared figures and/or tables, authored or reviewed drafts of the article, and approved the final draft.
- Nina Fefferman conceived and designed the experiments, authored or reviewed drafts of the article, and approved the final draft.
- L. Michael Romero conceived and designed the experiments, performed the experiments, analyzed the data, prepared figures and/or tables, authored or reviewed drafts of the article, and approved the final draft.

## Animal Ethics

The following information was supplied relating to ethical approvals (*i.e.*, approving body and any reference numbers):

All experiments were approved by the Tufts University Institutional Animal Care and Use Committee.

## Data Availability

The raw data are available in the Supplemental Files.

## Supplemental Information

Supplemental information for this article can be found online at http://dx.doi.org/10.7717/peerj.15661#supplemental-information.

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
