# Peer review of "Varying intensities of chronic stress induce inconsistent responses in weight and plasma metabolites in house sparrows (Passer domesticus)"

_PeerJ, doi:10.7717/peerj.15661_

## Round 0.1 · original submission · Major Revisions

This manuscript has been reviewed by two experts in the field. Both of them thought it is a well-designed work and a well-written manuscript. However, there is some still unclear issues regarding methods and explanation of the results, addressed by the reviewers.

Reviewer 1 ·

Excellent Review

This review has been rated excellent by staff (in the top 15% of reviews)
EDITOR COMMENT
The comments not only raise questions on the specific sections, but also explain why the audience will question that specific information. The comments also provide some unique viewpoints with specific evidence, which is greatly helpful for extending the current knowledge of the manuscript. Therefore, these responsible comments are very constructive and specific for improving the current manuscript. By having these comments, I believe the authors would be very goal-oriented and enjoy the process of improving their manuscript. We are thankful for the effort contributed by the responsible reviewer.

Basic reporting

The manuscript tackles long-standing issues that are of considerable interest in the field of stress physiology: What symptoms separate acute from chronic stress? What are metabolic changes that are associated with chronic stress? Are these changes proportional to the intensity and duration of stress? Are they reversible after cessation of chronic stress exposure? The study was conducted by an experienced and capable team, and the experimental model (House Sparrow) is appropriate, having been extensively used – including by the authors – for investigations related to stress physiology. The manuscript is nicely written and edited and has the potential to significantly contribute to the field. However, procedural aspects of the work are problematic and/or described in insufficient detail, damping this reviewer’s enthusiasm for the findings and their implications.

Experimental design

** Major issues:
L93: A major issue concerns the diet that captive birds received and the consequences of this diet.
First, the captive study lasted for approximately six months during which sparrows received only sunflower and millet seeds. On a chronic basis, this is a nutritionally very unbalanced diet which is deficient in several vitamins and likely other important nutrients, and this could in turn have affected the results. Why did the authors choose this diet as compared, e.g., to a commercial, nutritionally balanced diet designed for small bird maintenance in captivity, as routinely done in many other studies?
Second, the authors argue in fed sparrows that plasma uric acid can be used as a proxy for food consumption (L 93) based on the fact that sunflower seeds are particularly rich in uric acid (which is correct). Nothing is unfortunately more uncertain. For one, they did not quantify the individual proportion of consumed millet vs. sunflower seeds. If this proportion varied individually and/or over time, as was likely the case at least for some birds, the amount of uric acid ingested varied in parallel, thereby also influencing blood uric acid concentrations and, therefore, introducing a lot of “noise” in the data that could mask effects of chronic stress.
In addition, the idea that blood uricemia is a proxy for food consumption is based on the results of a previous study (Beatty et al. 2022). Here, fasted sparrows were bled, re-fed for one hour, and bled again. Re-feeding for one hour was associated with elevated uricemia relative to the fasted state. However, this experimental design did not separate the effect of re-feeding from that of the stress associated with handling and bleeding before food was provided. Thus, in Beatty et al. (2022) as well as here, the increase in blood uricemia (as well as in blood glycemia) could be acute stress- and not refeeding-related. The authors will likely (and justifiably) argue that several previous studies found decreased rather than increased blood uricemia in response to acute stress/elevated corticosterone. However, as original work on this subject indicates (Physiol. Biochem. Zool. 2008: 81/4 p. 463) the relationships between these variables is complex and variable. What is really needed, therefore, is a test comparing blood uricemia in re-fed sparrows that were or not stressed one hour before the second bleed. Include this information if available. This test is important also for another reason: The authors did not measure how much food sparrows ate when re-fed or for this matter, if they even ate during the hour preceding the second blood sample collection. The passage rate of food through the avian GI takes several hours (at least in chickens - it may take less time in small granivorous passerines; Poultry Sci. 1987: 66 p. 289). Even in the latter case, and following the authors’ assertion, one would need to assume that during the hour separating the two bleeds, sparrows did feed and the ingested food was at least partially digested and nutrients (incl. uric acid) absorbed into the blood in sufficient amount to elevate uricemia. It would seem critical in the manuscript to support these assertions with actual data.
Circling back to the birds’ diet and as I understand it, a primary objective of the work was to describe changes in blood uricemia (which is associated with protein catabolism) during chronic stress. This in and by itself should have oriented the authors toward selecting a diet that was less rich in uric acid than was the case, thereby facilitating the detection of protein catabolism-related changes in blood uricemia.

L150: Was each experimental group in it own room? The stress protocol involved (for example) playing radio in the bird room and a person speaking in the room. What procedure was implemented such that only the targeted group was exposed to each aspect of the stress protocol at a time?

L170: Each bird was weighed after the 2nd bleed, which itself followed one hour of birds having the opportunity to feed. Why were birds weighed then vs. just before re-feeding, which would have avoided the potential for food added to the GI (in unknown amount and not of interest in the context of the present study) influencing the birds’ weight?

** Minor issues:
L62… exerts major effects…

L116: [House sparrows] are easily brought into captivity: What is meant here? Do the authors imply that House Sparrows do not have to be coerced to be brought into captivity? Explain.

L123: … if a bird dropped below 85% of its pre-experiment weight…. When exactly was pre-experiment weight measured? At capture? After acclimation to captivity? Inform.

Validity of the findings

L125: By the end of Bout 6, 60% of the medium stress group birds were dead, and this percentage had increased by 70% by the end of the study. This is concerning because even though a Cox survival analysis was not significant, the fact is, for example, that by the end of Bout 6, mortality was nearly significantly higher in the medium stress group than in the low stress group (Fisher test: 0.057). Initial sample sizes were set based a solid power analysis (L 132), but the odds of detecting group differences decreased during the course of the study due to mortality and a consequent decrease in sample sizes. Thus, how much confidence did the authors have in detecting statistically significant group differences by the end of the study (say, Bout 5 and 6, and after recovery)? This should be documented, for example by including effect sizes and changes thereof that are associated with statistical comparisons during the course of the experiment. Providing the effect size that is associated with body masses at the end of the study would be particularly illuminating because at this point there was a lot of overlap between the low and high stress groups (Fig. 3C). The authors use considerable space to discuss differences between these groups and their importance. However, even though group differences are statistically significant, one has to wonder about their biological importance/relevance. Providing the associated effect size would help readers decide for themselves.

Reviewer 2 ·

Basic reporting

The paper is well written throughout. The background was well covered and the use of the literature appropriate. The length of the paper was good and the discussion was a fair representation of the work and the broader context of the literature.

Experimental design

The experiment is well designed and the authors have been careful in their description of methodology and analysis.
The results are well analysed and nicely presented, even though the results themselves were very inconsistent and they did not find the anticipated key difference between stress treatments. Nevertheless the data is worth publishing and will be a useful contribution to the field, in presenting a thorough set of data on blood levels of glucose, ketones and uric acid in relation to stress. The main conclusion of the study being that they are not consistently useful as measures of chronic stress.

Validity of the findings

In the study the authors used house sparrows to investigate the physiological transition from acute to chronic stress. The experiment is well designed and the authors have been careful in their description of methodology and analysis.
The findings are completely valid and the study is sound, with clear conclusions, albeit rather unsatisfying ones given all the work they have done.

Additional comments

Overall, a useful study that is well put together. I did not identify any issues that needed to be addressed.

---

## Round 0.2 · accepted · Accept

I am pleased to inform you that this manuscript is accepted.

Reviewer 1 ·

Basic reporting

Interesting study that follows previous research on the same topic by the same as well as other teams. The manuscript is well structured and written, and it refers to relevant published papers. It is prepared professionally and appropriate for publication in this journal.

Experimental design

No issue here. The questions and hypotheses are well defined, the study includes appropriate controls, and the method used to analyze data statistically is appropriate.

The methods are described in sufficient details that the study could be replicated.

Validity of the findings

The results are novel and expand on previous knowledge. The conclusions are sound albeit limited given that the effects of the treatments were variable and not entirely predictable.